# Comparison of Different Synthetic Routes of Hybrid Hematite-TiO_2_ Nanotubes-Based Electrodes

**DOI:** 10.3390/ma14164501

**Published:** 2021-08-11

**Authors:** Ana Luiza Fornazari, Judith Castillo-Rodriguez, Daniel Correa-Encalada, Enrique A. Dalchiele, Samuel A. Hevia, Rodrigo del Río

**Affiliations:** 1Departamento de Química Inorgánica, Facultad de Química y de Farmacia, Pontificia Universidad Católica de Chile, Santiago 7820244, Chile; adetoledo@uc.cl (A.L.F.); judycr89@gmail.com (J.C.-R.); dlcorrea@uc.cl (D.C.-E.); 2Centro Investigación en Nanotecnología y Materiales Avanzados UC (CIEN-UC), Pontificia Universidad Católica de Chile, Santiago 7820244, Chile; samuel.hevia@fis.uc.cl; 3Facultad de Ingeniería, Instituto de Física, Julio Herrera y Reissig 565, Montevideo 11000, Uruguay; dalchiel@hotmail.it; 4Instituto de Física, Facultad de Física, Pontificia Universidad Católica de Chile, Santiago 7820244, Chile

**Keywords:** TiO_2_ nanotubes, hematite, electrochemistry, SILAR, CVD

## Abstract

Nowadays, green hydrogen is an important niche of interest in which the search for a suitable composite material is indispensable. In this sense, titanium oxide nanotubes (TiO_2_ nanotube, TNTs) were prepared from double anodic oxidation of Ti foil in ethylene glycol electrolyte. The morphology of the nanotubes was characterized by scanning electron microscopy (SEM) and X-ray diffraction (XRD). Once characterized, nanotubes were used as templates for the deposition of hematite. The use of three synthetic procedures was assayed: Chemical Vapor Deposition (CVD), Successive Ionic Layer Adsorption and Reaction (SILAR), and electrochemical synthesis. In the first case, CVD, the deposition of hematite onto TiO_2_ yielded an uncovered substrate with the oxide and a negative shift of the flat band potential. On the other hand, the SILAR method yielded a considerable amount of hematite on the surface of nanotubes, leading to an obstruction of the tubes in most cases. Finally, with the electrochemical synthesis, the composite material obtained showed great control of the deposition, including the inner surface of the TNT. In addition, the impedance characterization showed a negative shift, indicating the changes of the interface electrode–electrolyte due to the modification with hematite. Finally, the screening of the methods showed the electrochemical synthesis as the best protocol for the desired material.

## 1. Introduction

Nowadays, nanoscience and nanotechnology are widely used due to their high potential to benefit several areas such as energy production, water decontamination, development of medicines, information and communication technologies, construction of more resistant materials, and light generation [1].

After discovering carbon nanotubes (CNTs) in the 1990s, other nanotubular shapes became of great interest to researchers due to their properties and excellent performances in various applications [2,3,4,5]. In this sense, the TiO_2_ nanotube (TNTs) and nanopore arrays are among the most favorable nanomaterials for research because they can be easily fabricated and exploited due to their large surface area. TNTs can be used in various applications [6], including fuel cells [7], photocatalytic systems, energy storage materials [8], gas and pH sensors [9], and agents for environmental analysis systems. The TNTs’ arrays are highly ordered and vertically oriented and have a large surface, low toxicity, photostability, and an excellent charge transfer property [10,11,12].

Specifically, TiO_2_ and iron oxide (hematite, α-Fe_2_O_3_) are two of the most studied materials for constructing photoanodes due to their resistance to photocorrosion, non-toxicity, natural abundance, and low-cost production [13]. However, some disadvantages, like their inherently low electrical conductivity and free limited medium path for the carriers’ diffusion, significantly affect TiO_2_ and α-Fe_2_O_3_ in photoelectrochemical water-splitting (PEC-WS) devices [5]. Nevertheless, it has been reported by Kment et al. [5] that in the interface of hematite-TiO_2_, some iron titanates’ compounds (Fe_2_TiO_5_ and others) are formed, showing an improvement in the photoelectrochemical performance. In general, combining these two materials into one composite is a widely studied path due to increased photocurrent and charge separation (despite the recombination in hematite) [14,15]. However, the importance of a 1D deposit in literature has been debated to overcome this system’s significant drawback: limited charge transfer [5]. In addition, our group has reported the substrate effect on the final material properties regarding similar systems [16]. Hence, the importance of performing a comprehensive study using the same conditions (i.e., TNT as a substrate) and different synthetic approaches to modify the TiO_2_ template with hematite. The study will lead to a better control of the composite’s final properties. A future application of the synthesized material in photoelectrochemical (PEC) water splitting for hydrogen production due to its cleanliness, inexpensiveness, and reliability [17] is expected. This work aimed to evaluate hematite deposition onto titanium nanotubes using different techniques to exploit the final material properties as a hybrid system for hydrogen generation.

## 2. Materials and Methods

For preparation of TiO_2_ nanotube arrays, highly ordered TiO_2_ nanotube arrays were synthesized by a double anodization process, reported in the literature [18]. In this regard, the first anodization process produces a disordered surface, while the second-step anodization creates a highly ordered surface [19]. All anodization experiments were carried out in a two-electrode electrochemical cell at room temperature. Ti foil and Pt foil were used as the anode and cathode, respectively. Before the electrochemical anodization, Ti foil (0.50 mm, 99.6% purity, Sigma Aldrich, Ñuñoa, Chile) was ultrasonically degreased in acetone, isopropanol, and methanol, for 15 min in each, and then chemically etched in a mixture of HF and HNO_3_ (HF:HNO_3_:H_2_O = 1:4:5 in volume, total 20 mL) for 1 min. Afterward, the foils were rinsed with deionized water and finally dried under a stream of nitrogen. The first-step anodization was performed under 60 V for 1 h in an ethyleneglycol solution containing 0.5 wt.% NH_4_F and 3 vol.% H_2_O. Then, the formed nanotube layer was removed by ultrasonication in deionized water for 1 min. The second-step anodization was carried out in the same electrolyte for 2 h. Finally, the prepared electrodes were rinsed with deionized water and dried at 60 °C for 3 h. The obtained TNT was used as substrate for the deposition of hematite using the following methods.

For the synthesis of α-Fe_2_O_3_, in the herein research, three synthetic approaches were followed to deposit iron oxide onto the previously synthesized TNT. Firstly, chemical vapor deposition was completed through thermal decomposition of ferrocene positioned at the beginning of the furnace, while the sample to modify was placed at 18 cm of the reservoir. The program used corresponded to an initial temperature of 200 °C with a heating ramp of 10 min (Ar 200 sccm), followed by a plateau at 200 °C for 10 with, then a heating ramp of 20 min to reach 500 °C (Ar 15 sccm) continued by the second plateau of 10 min (Ar 200 sccm). Then, it was completed a heating process of 30 min to reach 550 °C and a plateau of 120 min at that temperature under oxygen atmosphere (O_2_ 200 sccm). Secondly, α-Fe_2_O_3_ was deposited via the SILAR method. In this case, numerous combinations were carried out, as summarized in Table 1. As observed, two iron precursors were assayed, namely, FeCl_3_·6H_2_O and Fe(NO_3_)_3_·9H_2_O, while, for the precipitating agent, only NaOH was used (in all cases, both concentrations were kept constant, i.e., 0.1 mol·L^−1^). In addition, a solvent evaluation was performed using methanol and water while the number of cycles was optimized, comparing 10 and 50 cycles. A SILAR cycle consists of the immersion of the previously formed TNT in the Fe^2+^ precursor reservoir for the amount of time stated in Table 1 $(t_{Fe^{3+}}$), then dipping the substrate in the corresponding solvent to eliminate the excess of precursor (**t_solv_**), followed by an immersion of the substrate into the sodium hydroxide reservoir (**t_NaOH_**) for the amount of time declared in Table 1, and finally rising the substrate using the same solvent (**t_solv_**).

Finally, iron oxide thin films were electrochemically grown by potential cycling, as per Schrebler et al. (2007) [20]. A typical three-electrode electrochemical cell geometry was used, comprising a previously obtained TNT substrate, a steel plate, and Ag/AgCl (1 M) as working, counter, and reference electrodes. The electrodeposition procedure was performed at 50 °C in an aqueous solution of 5 mmol·L^−1^ FeCl_3_, 5 mmol·L^−1^ KF, 1 mol·L^−1^ H_2_O_2,_ and 0.1 mol·L^−1^ KCl. The electrochemical growth was completed using a potential cycling procedure at a potential sweep rate of 0.02 V·s^−1^, from 0.45 to −0.45 V for six cycles in total. The iron oxide thin films were annealed under an air atmosphere for 60 min at 500 °C in a tube furnace to increase crystallinity and eliminate organic residues.

For characterization, the electrochemical measurements and electrodepositions were carried out using a CH Instruments potentiostat/galvanostat, using a three-electrode system of Ag/AgCl (1 M) and Pt wire as a reference and counter-electrodes, respectively. Finally, in this configuration, the working electrode corresponded to the previously obtained and modified substrates, namely, TNT and TNT-Fe_2_O_3_ through all the methods reported. The electrochemical characterization (using Electrochemical Impedance Spectroscopy) was performed from −0.8 to 0.2 V, with an amplitude of 5 mV, 1000 Hz using NaOH (1 M). The graphs presented correspond to a Mott–Schottky treatment, as usually reported for semiconductors to evaluate the E_FB_ [21]. On the other hand, the morphological study was done using a field emission scanning electron microscope (FE-SEM, FEI Quanta 250). The length, inner diameter, and the wall thickness of the tubes were measured by segmenting SEM micrographs and using image analysis through ImageJ software. The cross section image was taken through a cut made with a scalpel, and the image was registered with a tilted sample holder to observe the tube. The elemental analysis was performed using energy-dispersive X-ray spectroscopy (EDAX, Materials Analysis Division, Model Octane Pro, Microscope Port Quanta 250 F). The structural characterization was performed in a Philips PW180 diffractometer (30 kV, 40 mA, CuKα radiation with λ = 1.5406 Å). Raman measurements were performed using a Witec Alpha 300 equipment.

The chemical characterization was conducted by X-ray photoelectron spectroscopy (XPS) with a FlexPS system from SPECS. The XPS was equipped with a hemispherical analyzer PHOBIOS 150, detector 1D-DLD, and monochromatic X-ray source FOCUS 500, providing Al Kα radiation with a characteristic energy of 1486.71 eV.

## 3. Results and Discussion

### 3.1. Surface Morphology of the Anodized TiO_2_ Nanotube Arrays

As can be seen in Equations (1) and (2), the generation of TiO_2_ nanotubes was based on two competing processes, the anodic formation of the oxide and its dissolution to form soluble fluoride complexes. The Ti^4+^ ions placed on the surface reacted with the dissolved O^2−^ in the electrolyte to form an oxide layer.
(1)$$\mathrm{Ti}+2H_{2}O\to \mathrm{Ti}O_{2}+4H^{+}+4e^{-}$$
(2)$$\mathrm{Ti}O_{2}+6F^{-}+4H^{+}\to TiF_{6}^{2-}+2H_{2}O$$

The increased surface area resulting from the formation of nanotubes improved the energy conversion efficiency of light by increasing the number of photogenerated electron-hole pairs [22].

During the synthesis, the low-current value observed at higher potential suggests a higher dissolution rate of TiO_2_ compared to its growth, resulting in a nanotubular structure as confirmed by the FE-SEM analysis in Figure 1a [20]. In this sense, the TNTs were circa 300 nm in diameter. No bundles, cracks, or over layers in the TNT films were observed. The formed TiO_2_ nanotubes electrodes may be combined with hematite generated by three different synthetic routes, leaving mixed compounds at the interface. The electrodes will have a highly illuminated surface area, and the incorporation of hematite will decrease the recombination of hollow-electron pairs due to the formation of a heterojunction [20]. In the future, the efficiency of these electrodes in the production of energy (production of H_2_ by water decomposition) can be evaluated.

The Raman spectra of the obtained nanotubes showed five signals corresponding to the anatase phase of TiO_2_ (even though the crystallinity can increase using a thermal treatment). In the literature, the peaks located at 144, 195, and 634 cm^−1^ are known as Eg peaks and are related to symmetric stretching vibrations of the bonds O-Ti-O in the structure [23]. The peaks located at 393 and 514 cm^−1^ represent B_1g_ and A_1g_ and are related to symmetric and antisymmetric vibrations of the bonds mentioned above. The nanotubes obtained were crystalline, as observed in Figure 1e with planes corresponding to anatase. In this case, diffraction peaks from Ti sheets were expected and observed as the most intense signals in the diffractogram, i.e., 40.2° (101) and 38.3° (002). The highest intensity peak regarding TiO_2_, as reported [24], corresponded to the peak (004), which indicated a strong orientation toward the direction 001 in the formed nanotubes. In addition, the lack of a peak at 27° or 31° indicated that there were no rutile or brookite impurities [23]. The crystallite size corresponded to 25 nm, using Scherrer’s equation. Finally, the electrochemical characterization through the Mott–Schottky plot indicated an n-type oxide with a flat band potential close to zero.

### 3.2. Hematite Deposition onto Anodized TiO_2_ Nanotube Arrays

Once the substrates were synthesized [18] and fully characterized, they were used as scaffolds for the deposition of hematite. In this sense, Figure 2 shows the results obtained for the deposition of hematite using three different methods.

In the following sections, a detailed characterization will be performed for each synthetic route to determine the better-suited modification of nt-TiO_2_ considering the high variability already reported regarding the impregnation method [21].

In the case of CVD, as observed in Figure 3, the presence of a slight deposition of hematite on the top of the nanotubes shows small islands of hematite and, as expected, the formation of a thin film over the previously formed nanotubes without essential changes in the morphology of the obtained material.

For the Raman spectrum, in which the main peak remained at the same position (144 cm^−1^), the lack of change in the spectrum was related to the intensity of the TNT peaks in the sample compared to hematite. The same scenario was observed in the XRD, in which the diffraction pattern in Figure 3c obtained for this composite material featured two critical aspects to be discussed. Firstly, it is important to notice the lack of hematite phase peaks that, as explained earlier, can be directly related to the amount of material deposited (low amount of hematite in the surface of nt-TiO_2_). Secondly, in the diffractogram, peaks at 25.3° (101) and 37.84° (004) indicate the presence of anatase [24]. The position shift of such peaks can be due to a substitution reaction at/near the nanotube’s surface during calcination. In this sense, the incorporation of Fe^3+^ would increase the crystal radius and shift the Bragg angle, as explained by M. Sołtys-Mróz et al., 2020 [21]. Finally, in the Mott–Schottky plot, a shift of the E_FB_ toward negative potentials indicates a change in the surface of the nt-TiO_2_ and charge transfer processes across the material, as discussed elsewhere [21]. Additionally, it has been previously reported as being due to increasing the film thickness and morphology of the deposit [25]. In addition, the graphs indicate a difference in the composition of the surface throughout the material, meaning that the modification was not as homogeneous as expected.

In the case of the SILAR method, several variables were assayed. First, it was studied the immersion time using nt-TiO_2_/α-Fe_2_O_3_ (t10), nt-TiO_2_/α-Fe_2_O_3_ (t60), and nt-TiO_2_/α-Fe_2_O_3_ (t180) with 10, 60, and 180 s, respectively. In this case, as observed in Figure 4, the morphology of the iron oxide deposition changed with time. Although there was a wider deposition for 180 s, most of the pores were clogged, which prevented the use of the inner area and led to a smaller photoactivity. In the case of different solvents, only water was successful in forming iron oxide.

Even though the literature has reported the synthesis of hematite in methanol [26,27], only the aqueous solution yielded the expected oxide. If compared to Figure 5a (nt-TiO_2_/α-Fe_2_O_3_ (H_2_O)) and Figure 5c (TiO_2_/α-Fe_2_O_3_ (MetOH)), low coverage of the nanotubes was obtained for methanol. In this case, the effect was stressed out using 10 and 50 cycles for nt-TiO_2_/α-Fe_2_O_3_ (H_2_O) and TiO_2_/α-Fe_2_O_3_ (MetOH), respectively; and not even using 50 cycles, the deposition of iron oxide was observed. In addition, the EDS measurements observed for both conditions showed the presence of Fe in the samples (Figure 5b,d). On the other hand, once water was selected as the solvent, the evaluation of the number of cycles was performed comparing nt-TiO_2_/α-Fe_2_O_3_ (10 cycles) (Figure 5e) and nt-TiO_2_/α-Fe_2_O_3_ (50 cycles) (Figure 5a).

As reported elsewhere, increasing the number of cycles led to an increase in the deposition of the oxide on the working surface. As observed, the use of 50 cycles provoked a high coverage. Hence, an obstruction of most of the pores was observed, decreasing the exposed area, as previously explained. Finally, the last aspect assessed was related to using different counterions: Cl^−^ (Figure 5e) and NO_3_^−^ (Figure 5f). In this case, the use of different salts played a vital role in the formation of hematite. As Baumler et al. (2019) [28] reported, for iron nitrate and chloride, a process of complex formation determines the hydration structure and symmetry of the final solution. These differences yield different deposition, with the total disappearance of the nanotubes, for the case of iron nitrate (Figure 5f). Although the different conditions were assayed, a partial conclusion on the SILAR approach was that numerous variables and conditions can interfere with the “ideal” (considering the goals of this research) deposition in this synthetic route.

Finally, the electrochemical deposition was evaluated. In this case, as observed in Figure 6, the technique rendered a more precise deposition, considering the outer surface of the nanotubes, as observed in the previous routes and the inner cavities of the tubes. The EDS measurement indicated iron and the rest of the components in small amounts, as expected. Raman spectra showed the same behavior and no changes in the intensity or the peaks’ number or shape. On the other hand, the diffraction pattern observed in Figure 6c, showed the same features observed above for the TNT with non-distinctive peaks of hematite.

Finally, in the Mott–Schottky plot, an E_FB_ of −0.2 V vs. Ag/AgCl indicated a modification of the surface states and, if compared to the previous methods, a more controlled surface composition was observed. It has been reported that polycrystalline electrodes, such as the studied composite material, have different crystallographic planes in contact with the electrolyte leading to the system’s different conductivity, leading to a change in the E_FB_. [29].

XPS characterization was performed on selected samples to ensure the presence of Fe_2_O_3_ on the TNT surface. The spectrum obtained as a survey (Figure 7a) showed the presence of Fe (high-resolution XPS of Fe2p in Figure 7b), Ti (high-resolution XPS of Ti2p in Figure 7c), and O (high-resolution XPS of O1s in Figure 7d), as reported elsewhere. In Ti2p, the split between the two peaks (1/2 and 3/2) corresponds 5.6 eV, which indicated the presence of TiO_2_ [15]. For Fe2p, the position of Fe2p 1/2 (724.2 eV) and Fe2p 3/2 (710.5 eV) were according to the formation of hematite [15]. Additionally, the satellite at 718.3 eV in all samples indicated the presence of α-Fe_2_O_3_ [15,30]. Only in the case of SILAR did a component at smaller energies for the Fe2p 3/2 appear, specifically at around 709.6 eV, which may indicate the formation of FeO using this method [31,32].

## 4. Conclusions

In the research herein, a thorough evaluation of the surface and inner modification of the tubes was performed. To this purpose, a three-method evaluation was completed. Firstly, in the CVD method, although it allows a homogenous deposition of the surface, the electrochemistry of the interface indicated the contribution of different surfaces (i.e., TiO_2_ and Fe_2_O_3_) and the deposition was located only at the top of the nanotubes.

Secondly, the SILAR method obtained a higher amount of hematite despite numerous variables to be considered. In this sense, it is mandatory to complete a much more thorough study of the system to understand the synthetic approach. In addition, using this method, FeO was probably formed, as observed from the XPS analysis. Finally, the electrochemical method provided an inner modification and an E_FB_ that was toward the negative potentials compared to the TiO_2_ and a neater plot. Overall, the better quality of the composite material obtained in the last approach foresees a successful future application in hydrogen generation.

## Figures and Tables

**Figure 1 materials-14-04501-f001:**
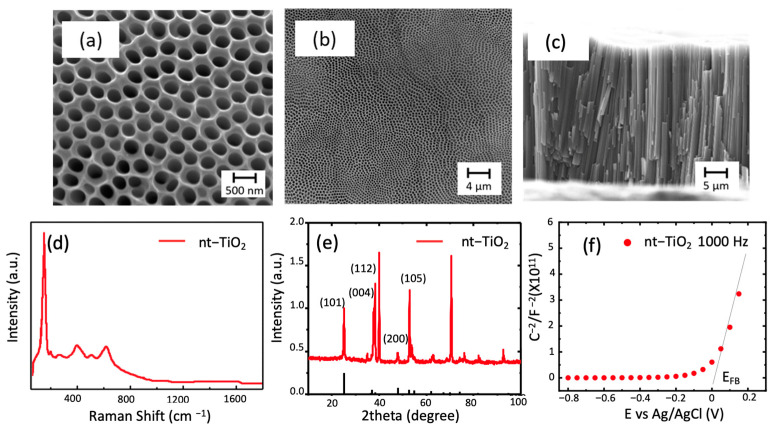
SEM micrographs of the top view of the TiO_2_ nanotubes formed by two anodization of titanium foil at 60 V with magnifications of (**a**) 200.000× and (**b**) 26.000×; (**c**) the cross section showing the length of the tubes with magnification of 13.000×; (**d**) Raman spectra of typical nt-TiO_2_ obtained; (**e**) diffraction spectra of the nanotubes (in this, the expected position for anatase is observed as black lines) (PDF 1-562); (**f**) Mott–Schottky plot of the sample.

**Figure 2 materials-14-04501-f002:**
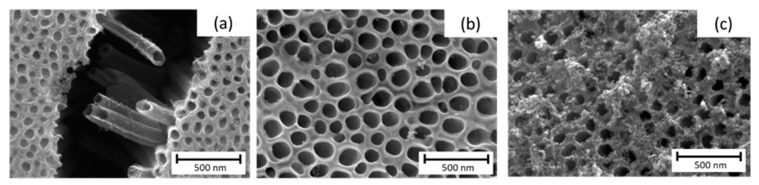
SEM micrographs of the top view of the nt-TiO_2_ /α-Fe_2_O_3_ formed using (**a**) electrochemical method, (**b**) CVD, (**c**) SILAR method.

**Figure 3 materials-14-04501-f003:**
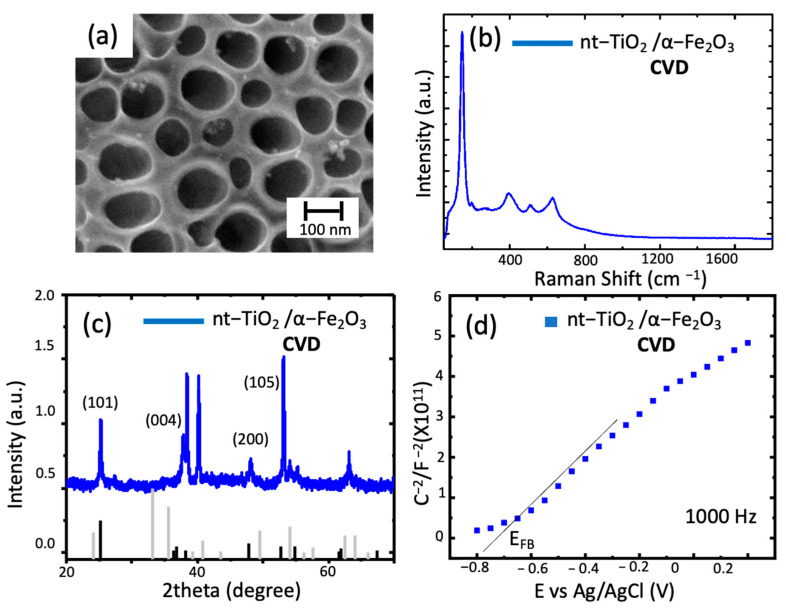
Characterizations of nt−TiO_2_/ α−Fe_2_O_3_. (**a**) SEM micrographs of the top view of the TiO_2_ nanotubes modified with hematite through CVD; (**b**) Raman spectra nt-TiO_2_/ α-Fe_2_O_3_ obtained; (**c**) diffraction spectra of the nanotubes (black lines correspond to TiO_2_, gray lines correspond to hematite); (**d**) Mott–Schottky plot of the sample at 1000 Hz.

**Figure 4 materials-14-04501-f004:**
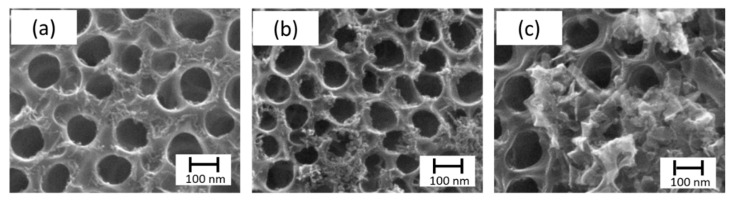
SEM micrographs of SILAR method. Top view of nt-TiO_2_ /α-Fe_2_O_3_, namely, (**a**) nt-TiO_2_/α-Fe_2_O_3_ (t10), (**b**) nt-TiO_2_/α-Fe_2_O_3_ (t60), (**c**) nt-TiO_2_/α-Fe_2_O_3_ (t180) with 10, 60, and 180 s, respectively.

**Figure 5 materials-14-04501-f005:**
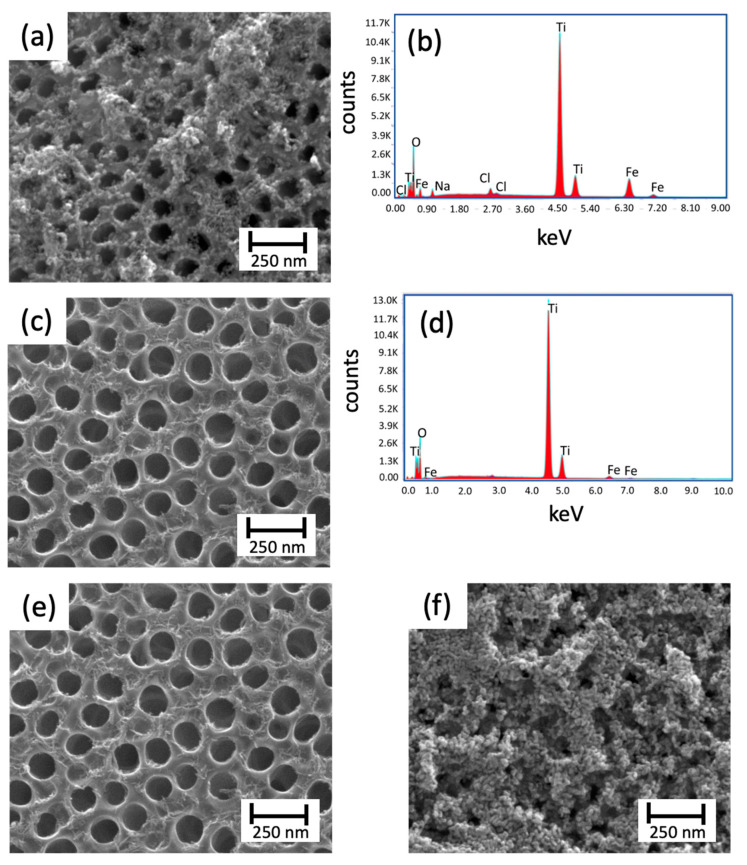
Top view SEM micrographs and EDS measurements of the TiO_2_ nanotubes modified with hematite through SILAR; (**a**,**b**) nt-TiO_2_/α-Fe_2_O_3_ (H_2_O); (**c**,**d**) nt-TiO_2_/α-Fe_2_O_3_ (MetOH) using 10 and 50 cycles, respectively; (**e**) nt-TiO_2_/α-Fe_2_O_3_ (Cl^−^); and (**f**) nt-TiO_2_/α-Fe_2_O_3_ (N$O_{3}^{-}$).

**Figure 6 materials-14-04501-f006:**
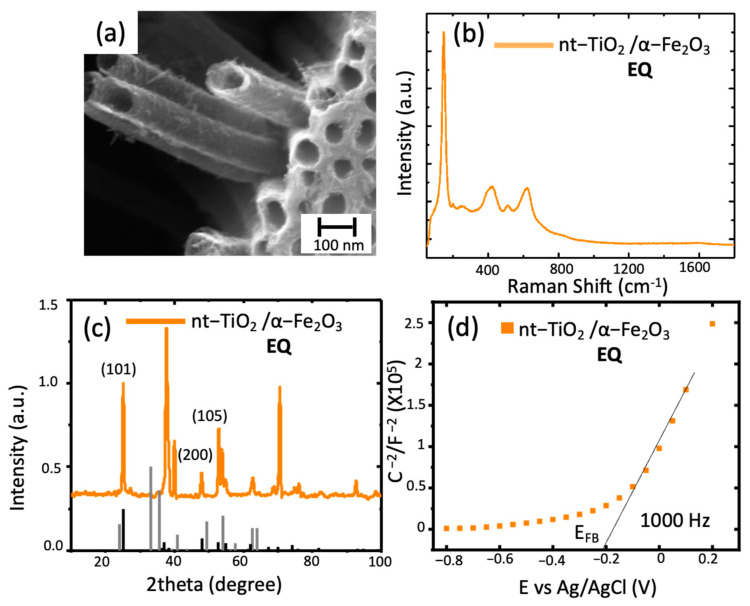
TiO_2_ nanotubes modified with hematite through electrochemical deposition: (**a**) SEM image of nt-TiO_2_/α-Fe_2_O_3;_ (**b**) Raman spectra nt-TiO_2_/ α-Fe_2_O_3_ obtained; (**c**) diffraction spectra of the modified nanotubes (black line corresponds to hematite PDF 1-1053); (**d**) Mott–Schottky plot of the sample at 1000 Hz.

**Figure 7 materials-14-04501-f007:**
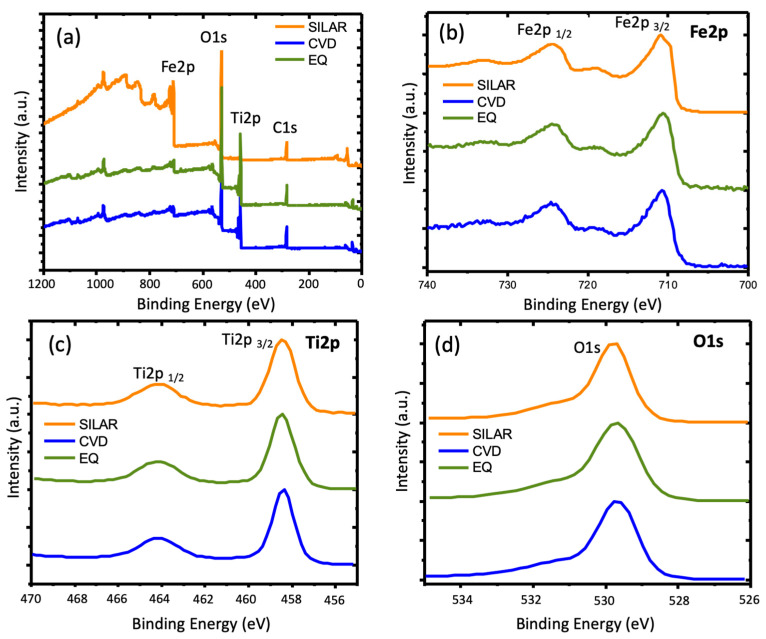
TiO_2_ nanotubes modified with hematite. (**a**) Survey spectrum, (**b**) Fe2p, (**c**) Ti2p, and (**d**) O1s of synthesized materials’ nt-TiO_2_/α-Fe_2_O_3_ using SILAR, CVD, and EQ as synthetic methods.

**Table 1 materials-14-04501-t001:** SILAR conditions assayed in the synthesis of α-Fe_2_O_3_ onto nt-TiO_2_. The parameters are as follows: ***t*** (***s***) corresponds to the time immersed in the corresponding solution (Fe^3+^, solvent used, and NaOH).

Sample	$t_{Fe^{3+}}$	t_solv_	t_NaOH_	t_solv_	No. of Cycles
nt-TiO_2_/α-Fe_2_O_3_ (t10)	10	10	10	10	10
nt-TiO_2_/α-Fe_2_O_3_ (t60)	60	10	60	10	10
nt-TiO_2_/α-Fe_2_O_3_ (t180)	180	10	180	10	10
nt-TiO_2_/α-Fe_2_O_3_ (H_2_O)	10	10	10	10	10
nt-TiO_2_/α-Fe_2_O_3_ (MetOH)	10	10	10	10	10
nt-TiO_2_/α-Fe_2_O_3_ (10 cycles)	10	10	10	10	10
nt-TiO_2_/α-Fe_2_O_3_ (50 cycles)	10	10	10	10	50
nt-TiO_2_/α-Fe_2_O_3_ (Cl^−^)	10	10	10	10	50
nt-TiO_2_/α-Fe_2_O_3_ (N$O_{3}^{-}$)	10	10	10	10	50

## Data Availability

A. Fornazari is the depositary of all the data generated by the study.

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
