# Peer review of "Comparison of Different Synthetic Routes of Hybrid Hematite-TiO2 Nanotubes-Based Electrodes"

_materials, 2021, doi:10.3390/ma14164501_

Round 1
Reviewer 1 Report
The present research paper describes and compare three different procedures for the functionalization of titania nanotubes with hematite. The samples are well characterized, and the manuscript is clear. In my opinion, there are only a few improvements that the authors should include in the text.
Line 18 a full stop is missing “…(XRD) Once…”
Line 27 the sentence is not complete “the impedance characterization showed a negative shift indicating…?”
Lines 55-67: The double anodization is well described and the reference is included but a brief sentence to explain why the procedure is repeated two times should be included.
Line 55, 91: a space is missing between numerical value and its unit.
Line 104: “the wall thickness of the tubes were measured by segmenting SEM micrographs”. Can you describe in the text how the cross section was realized? Sometimes the wrong procedure can bring to artefacts in the SEM analysis.
Figure 5 c d and insets captions are missing. Regarding these images also in the text the description is not very clear. If I understand well Figure 5a is H2O+Cl- and 5d MeOH+NO3-, but what about figure 5b and 5c?
Raman, XRD and electrochemical characterization of SILAR deposition is missing, why didn’t you not include them?
Author Response
We sincerely appreciate all comments related to the MS. In the following letter we, the authors, address your inquiries:
Line 18 a full stop is missing “…(XRD) Once…”
Corrected, the full stop was included. The MS was revised to avoid other typographical errors.
Line 27 the sentence is not complete “the impedance characterization showed a negative shift indicating…?”
Corrected, the sentence was complete to explain the changes due to surface modification
Lines 55-67: The double anodization is well described and the reference is included but a brief sentence to explain why the procedure is repeated two times should be included.
A brief comment about the doble anodization procedure was included, explaining that the first layer was removed to obtain well aligned nanotubes in the second one.
Line 55, 91: a space is missing between numerical value and its unit.
This comment was corrected
Line 104: “the wall thickness of the tubes were measured by segmenting SEM micrographs”. Can you describe in the text how the cross section was realized? Sometimes the wrong procedure can bring to artefacts in the SEM analysis.
A comment about the cross-section image was included as requested
Figure 5 c d and insets captions are missing. Regarding these images also in the text the description is not very clear. If I understand well Figure 5a is H2O+Cl- and 5d MeOH+NO3-, but what about figure 5b and 5c?
The caption and the main text were corrected as requested
Raman, XRD and electrochemical characterization of SILAR deposition is missing, why didn’t you not include them?
The authors completely understand the importance of a more thorough characterization, specifically Raman and XRD of the samples obtained through SILAR deposition. In this sense, some characterizations were partially done because they were completed during March 2020, and the lockdown did not allow to complete all of them correctly. This revised version of the MS includes other characterizations performed this month to support the herein conclusions.

Reviewer 2 Report
The manuscript contains serious flaws and erros. Sometimes sentences are incomplete or informations are missing. Some experimental methods used are not described. Images are of low quality and should be better explained. The discussion should be significantly improved and be much deeper. Many references are missing. In addition, style and english are not acceptable.
The work cannot be further considered for publication in Materials.
Author Response
We would like to thank you for your time and consideration. As authors of the herein MS, we consider the stated opinion of the referee, and we will explain in the following letter the main reasons for the MS publication according to our criteria.
The manuscript contains serious flaws and erros. Sometimes sentences are incomplete or informations are missing. Some experimental methods used are not described. Images are of low quality and should be better explained. The discussion should be significantly improved and be much deeper. Many references are missing. In addition, style and english are not acceptable.
The work cannot be further considered for publication in Materials.
First of all, we would like to apologize if the herein MS contains errors. For this new version of the MS we have worked to correct such flaws. Considering the objective of the presented research, we can mention that Kment et.al. (Chem.Soc. Rev. 2017, 46, 3716) discuss that in the interface of hematite-TiO2, some iron titanates compounds (Fe2TiO5 and others) are formed, and they show an improvement in the photoelectrochemical performance. Based on these findings, we initially tried to show if the synthetic method affects this performance (through changes in the interface). During the study, the sanitary condition in the country changed, and until now, there are numerous restrictions to use the research facilities located at the university. Considering this new scenario, we changed the focus of the present investigation to show how the synthetic route could have differences (morphologically and structurally) that will, in the future, lead to differences in the photoelectrocatalytic performance.

Reviewer 3 Report
The manuscript provides all comprehensive information for scientist and other readers interested in differences in characteristic of TiO2 NT hybrid material, obtained by various method. Discussion is in experimental data are in accordance with experimental data. Used techniques for the characterization of prepared materials are well chosen.
Author Response
We sincerely appreciate the feedback related to the MS, and in the following letter, we will try to respond to the comments:
The manuscript provides all comprehensive information for scientist and other readers interested in differences in characteristic of TiO2 NT hybrid material, obtained by various method. Discussion is in experimental data are in accordance with experimental data. Used techniques for the characterization of prepared materials are well chosen.
After the revision, we enhanced the references used, improved some figures' quality, and added extra characterizations. So, we kindly appreciate the evaluation.

Reviewer 4 Report
The authors present a comparitive analysis of three different methods for coating titania nanotube coated titanium electrodes with hematite.
Overall I think this manuscript is of a very low quality and I believe that it is not suitable for publication in Materials. I have a number of major concerns which have led me to this:
1) The authors say the aim is to identify the best method to produce the materials to make hydrogen...but how do they know what they have made would be good for this when they don't test it for that purpose?
2) The introduction is incredibly low on details and contains only 6 references?
3) What about this work is novel? I'm not clear any of it is?
4) There is a lack of detail in the experimental (some of the techniques used aren't even mentioned!) that would prevent anyone from attempting to repeat this work.
5) The authors show no evidence that any Fe containing product was even deposited by two of the methods, and the one that does definitely have some Fe deposited shows no evidence that it was hematite? The paper refers to hematite in the title but there is no evidence at any point that it was made.
In addition the following minor issues are a cause for concern.
1) Generally the quality of the writing must be improved as it is unclear, imprecise and full of spelling mistakes.
2) Why does the method used produce nanotubes?
3) Data presentation is poor, peaks are not labelled or appropriately compared. This is particularly true for the XRD.
4) The XRD data is presented very poorly, why are the intensities so wildly different between the samples and why are the baselines so thick?
5) Figures are presented that are never referred to in the text or sometimes even within their own Figure heading?
Author Response
We sincerely appreciate the comments regarding the MS and in the following letter we will try to respond your inquiries:
The authors present a comparitive analysis of three different methods for coating titania nanotube coated titanium electrodes with hematite.
Overall I think this manuscript is of a very low quality and I believe that it is not suitable for publication in Materials. I have a number of major concerns which have led me to this:
1) The authors say the aim is to identify the best method to produce the materials to make hydrogen...but how do they know what they have made would be good for this when they don't test it for that purpose?
We understood that without a photoelectrocatalytic test or photocurrent we don’t have measurable information about the performance of the materials obtained. However, from literature we can expect to obtain high photocurrent and photoactivity in general for this composite material. In addition, since March 2020 our research facilities have remain closed due toa national lockdown. Just now we are opening partially the laboratories, with some restrictions. So, we think that by the moment we can show the differences in the material obtained and from our results the electrochemical method shows the best performance. In this sense, the 1D structure will be fully covered but not clogged, which will lead to increase the surface area.
2) The introduction is incredibly low on details and contains only 6 references?
We kindly appreciate this comment and in this sense some references were added as well as information in general to improve the quality of the introduction with more details.
3) What about this work is novel? I'm not clear any of it is?
This work aimed to evaluate, if the synthetic methods affect the performance of the material applied to the water splitting process. This was thinking considering that the interface TiO2-Hematite could affect the performance, specifically iron titanates, such as, Fe2TiO5(Kment et.al., Chem.Soc. Rev. 2017, 46, 3716). At first, the electrochemical method improves the surface area modified, covered all the tubes without clogging, but the final test was not complete due lockdown. It is our believe, that in the meantime we can show the difference (morphologically and structurally) in the material obtained considering three methods in one manuscript using the same type of substrate. From our results (using other systems, Castillo-Rodirguez et. al J Solid State Electrochem 24, 797–808 (2020)), the use of different substrates can lead to a wide variation in the final deposit. Hence, having the same template and using different methods lead us to think that the electrochemical method will have the best performance once the photocurrent and photochemical measurements are done.
4) There is a lack of detail in the experimental (some of the techniques used aren't even mentioned!) that would prevent anyone from attempting to repeat this work.
The experimental section was improved to show more details and some characterization techniques was added to obtain a better characterization of the materials obtained.
5) The authors show no evidence that any Fe containing product was even deposited by two of the methods, and the one that does definitely have some Fe deposited shows no evidence that it was hematite? The paper refers to hematite in the title but there is no evidence at any point that it was made.
To show the presence of hematite, XPS was registered to selected samples of all the methods assayed. In the showed spectra the presence of hematite was detected. .
In addition the following minor issues are a cause for concern.
1) Generally the quality of the writing must be improved as it is unclear, imprecise and full of spelling mistakes.
Some errors were corrected to improve the quality of the MS.
2) Why does the method used produce nanotubes?
The formation of nanotube arrays in fluoride containing electrolytes is the result of three simultaneously occurring processes: (1) field assisted oxidation of Ti metal to form titanium dioxide, (2) field assisted dissolution of Ti metal ions in the electrolyte, and (3) chemical dissolution of Ti and TiO2 due to etching by fluoride ions. In general, it appears that the key to successfully achieving very long nanotube arrays is to minimize the water content in the anodization bath (Karthik Shankar et al 2007 Nanotechnology 18 065707)
3) Data presentation is poor, peaks are not labelled or appropriately compared. This is particularly true for the XRD.
The figures were redrawn to show better definition of the results obtained and some signals were marked. In the case of XRD, the figures were normalized.
4) The XRD data is presented very poorly, why are the intensities so wildly different between the samples and why are the baselines so thick?
In our results this can be explained considering that the thick baseline can be due to both a very thin film and/or a nanocrystalline material with very small crystallite sizes. This two facts produces more noised XRD signals. Specially in the case of the electrochemical method.
5) Figures are presented that are never referred to in the text or sometimes even within their own Figure heading?
This was improved and checked that all the figures are cited adequately in the text.

Reviewer 5 Report
The authors present a study about the deposition of Fe2O3 on TiO2 NTs. Fe2O3 is deposited by SILAR, CVD, and electrochemical deposition.
The research presented is not new and other papers dealing with the coupling between TiO2 and Fe2O3 are present in the literature.
I have some observations and comments:
1 - The authors should explain clearly the reason why the deposition of Fe2O3 on TiO2 should improve the photoelectrochemical performances of TiO2. Are the expecting to have a doping or a junction between Fe2O3 and TiO2?
2 - It is not clear from the experimental procedure if after SILAR and electrochemical deposition the samples are annealed.
3 - The procedure used for the electrochemical deposition of Fe2O3 is not well described. A reference is missing and the reaction that leads to Fe2O3 deposition should be written.
4 - Page 3, lines 132 - 135: "incorporation of hematite will decrease the recombination .." A reference is missing and a clear explanation of what expected should be given.
5 - The authors say that small islands of hematite are visible in Figure 3a. I actually don't see anything excluding 3 tiny particles on the edge of a pore. The presence of Fe2O3 should be checked by XPS or by a simple chemical analysis.
6 - Page 5, line 186: "the shift in the position of these peaks can be due .." How much is the shift?
No clear evidence of Fe2O3 deposition is ever shown. The only thing that changes is the MS plot. The authors should at least report XPS, UV-Vis spectra (Diffuse reflectance), and the photoelectrochemical performances (photocurrent).
The study is not complete.
Author Response
We appreciate the comments related to the MS and we will try to respond the enquiries in the following letter:
The authors present a study about the deposition of Fe2O3 on TiO2 NTs. Fe2O3 is deposited by SILAR, CVD, and electrochemical deposition.
The research presented is not new and other papers dealing with the coupling between TiO2 and Fe2O3 are present in the literature.
Although the referee is correct because in the literature numerous papers are related to this issue, we try to show how the different synthetic methods produce different final materials regarding the surface area and other issues. Our focus is to improve the final composite performance regarding hydrogen production, especially considering the formation in the interface of Fe2TiO5 and other compounds ((Kment et.al., Chem.Soc. Rev. 2017, 46, 3716). Due to the restriction of access to the laboratories since March 2020, specially Mach 2021, we have not tested this last issue. So, we show in the MS the difference observed regarding these three methods.
I have some observations and comments:
1 - The authors should explain clearly the reason why the deposition of Fe2O3 on TiO2 should improve the photoelectrochemical performances of TiO2. Are the expecting to have a doping or a junction between Fe2O3 and TiO2?
According to the work of Kment et.al. (Chem.Soc. Rev. 2017, 46, 3716), the relative misalignment of hematite and TiO2 band edges can be exploited to design a suitable heterojunction for enhancing the charge carrier separation and specific conditions. Also, the formation of Fe2TiO5, Fe3TiO4, and FeTiO3 in the interface are expected to exhibit remarkable performance in the water splitting catalysis.
2 - It is not clear from the experimental procedure if after SILAR and electrochemical deposition the samples are annealed.
A comment explained was included
3 - The procedure used for the electrochemical deposition of Fe2O3 is not well described. A reference is missing and the reaction that leads to Fe2O3 deposition should be written.
The experimental section was improved to clarify all the methods used. However, in a previous paper, we presented the electrochemical synthesis of hematite, in the herein MS the same procedure was used to modify the TiO2. (Schrebler et. al. Eletrochem. Solid-State Let. 9 (2006) C110).
4 - Page 3, lines 132 - 135: "incorporation of hematite will decrease the recombination .." A reference is missing and a clear explanation of what expected should be given.
Such was mentioned, hematite and specifically the interface TiO2-hematite help in the charge carrier separation (Kment et.al., Chem.Soc. Rev. 2017, 46, 3716). (Bhagya, T.C.; et. Photochem. Photobiol. Sci. 2019, 18)
5 - The authors say that small islands of hematite are visible in Figure 3a. I actually don't see anything excluding 3 tiny particles on the edge of a pore. The presence of Fe2O3 should be checked by XPS or by a simple chemical analysis.
XPS were performed in order to improve the characterization of the material.
6 - Page 5, line 186: "the shift in the position of these peaks can be due .." How much is the shift?
This was included
No clear evidence of Fe2O3 deposition is ever shown. The only thing that changes is the MS plot. The authors should at least report XPS, UV-Vis spectra (Diffuse reflectance), and the photoelectrochemical performances (photocurrent).
Now the XPS is included, and it shows the formation of hematite as expected.
The study is not complete.
5) Figures are presented that are never referred to in the text or sometimes even within their own Figure heading?
This was corrected.

Round 2
Reviewer 2 Report
The authors improved the manuscript significantly.
The following points should be further considered:
line 172: "The nanotubes ob-172 tained were crystalline" - can the author describe the crystalline character more in detail (e.g. texture coefficients, crystallite size, influence of strain)
In addition, Fig. 3 and 6 contain no markers for the diffraction peaks.
line 265: "XPS characterization was performed on selected samples to ensure the presence of Fe2O3 265 on the TNT surface."
XPS is sensitive to the surface. How the composition changes with depth?
Other comments:
line 118:electrochemical characterization (EIS) - EIS means Electrical Impedance Spectroscopy
line 134:Al ka radiation - Al Kα
image quality of EDX-spectra can be improved
Author Response
We would like to thank you for your time and consideration in the first revision. As authors of the herein MS, we consider the stated opinion of the referee, and we will explain in the following letter the last changes:
The authors improved the manuscript significantly.
The following points should be further considered:
line 172: "The nanotubes ob-172 tained were crystalline" - can the author describe the crystalline character more in detail (e.g. texture coefficients, crystallite size, influence of strain)
In addition, Fig. 3 and 6 contain no markers for the diffraction peaks.
The crystallite size of the TNT was included and the markers of the crystallographic planes were highlighted in the figures. In addition, the XRD data of the hematite were added.
line 265: "XPS characterization was performed on selected samples to ensure the presence of Fe2O3 265 on the TNT surface."
XPS is sensitive to the surface. How the composition changes with depth?
The referee has a point considering that XPS is a surface sensitive technique. In the cases of CVD and Silar the materials obtained could present differences in the composition, but at this moment we did not performed any study in this sense. On the other hand, the electrochemical synthesis has shown a more homogeneus and thin deposit (such as shown in the SEM images). So, it is our believe that that little differences in composition could be present in this promising material
Other comments:
line 118:electrochemical characterization (EIS) - EIS means Electrical Impedance Spectroscopy
The characterization with Electrochemicla Impadance Spectroscopy was rewrited to a more clear lecture
line 134:Al ka radiation - Al Kα
This was corrected
image quality of EDX-spectra can be improved
The image was redrawn in order to show in a best quality the EDX

Reviewer 4 Report
In this revision the authors have done a commendable job in addressing my concerns with the original manuscript, I am particularly happy with the addition of the XPS.
I am now satisfied that there is something novel to this work, but I would like the authors to make this point about the novelty more clear in the manuscript as well (rather than just in their response to me).
Overall this manuscript is significantly improved and subject to the minor addition requested above I think it may be suitable for publication in Materials.
Author Response
We sincerely appreciate the comments regarding the MS in the revised version and in the following letter we will try to respond your inquiries:
In this revision the authors have done a commendable job in addressing my concerns with the original manuscript, I am particularly happy with the addition of the XPS.
I am now satisfied that there is something novel to this work, but I would like the authors to make this point about the novelty more clear in the manuscript as well (rather than just in their response to me).
In the introduction, the effects of the FexTiOx compounds were included in order to show how the synthetic method could affect the final composite.
Overall this manuscript is significantly improved and subject to the minor addition requested above I think it may be suitable for publication in Materials.

Reviewer 5 Report
The authors have added important data like the xps spectra. These new data are more or less in tune with the rest of the characterizations. The differences in the MS plots are surprising and eventually should be further discussed.
Author Response
We appreciate the comments related to the revised version of our MS and we will try to respond the enquiries in the following letter:
The authors have added important data like the xps spectra. These new data are more or less in tune with the rest of the characterizations. The differences in the MS plots are surprising and eventually should be further discussed.
A comment of the Efb was included
